# Detection of *Pneumocystis* and Morphological Description of Fungal Distribution and Severity of Infection in Thirty-Six Mammal Species

**DOI:** 10.3390/jof9020220

**Published:** 2023-02-07

**Authors:** Christiane Weissenbacher-Lang, Barbara Blasi, Patricia Bauer, Diana Binanti, Karin Bittermann, Lara Ergin, Carmen Högler, Tanja Högler, Magdalena Klier, Julia Matt, Nora Nedorost, Serenella Silvestri, Daniela Stixenberger, Liang Ma, Ousmane H. Cissé, Joseph A. Kovacs, Amélie Desvars-Larrive, Annika Posautz, Herbert Weissenböck

**Affiliations:** 1Department for Pathobiology, Institute of Pathology, University of Veterinary Medicine Vienna, Veterinärplatz 1, 1210 Vienna, Austria; 2Critical Care Medicine Department, NIH Clinical Center, National Institutes of Health (NIH), 10 Center Drive, Bethesda, MD 20892, USA; 3Department for Farm Animals and Veterinary Public Health, Institute of Food Safety, Food Technology and Veterinary Public Health, University of Veterinary Medicine Vienna, Veterinärplatz 1, 1210 Vienna, Austria; 4Complexity Science Hub Vienna, Josefstädter Straße 39, 1080 Vienna, Austria; 5Department for Interdisciplinary Life Sciences, Research Institute of Wildlife Ecology, University of Veterinary Medicine Vienna, Veterinärplatz 1, 1210 Vienna, Austria

**Keywords:** *Pneumocystis* species, in situ hybridization, lung histopathology, Artiodactyla, Carnivora, Chiroptera, Eulipotyphla, Perissodactyla, primates, Glires

## Abstract

*Pneumocystis* spp. are thought to adapt to the lungs of potentially all mammals. However, the full host range, fungal burden and severity of infection are unknown for many species. In this study, lung tissue samples originating from 845 animals of 31 different families of eight mammal orders were screened by in situ hybridization (ISH) using a universal 18S rRNA probe for *Pneumocystis*, followed by hematoxylin and eosin (H&E) staining for determining histopathological lesions. A total of 216 (26%) samples were positive for *Pneumocystis* spp., encompassing 36 of 98 investigated mammal species, with 17 of them being described for the first time for the presence of *Pneumocystis* spp. The prevalence of *Pneumocystis* spp. as assessed by ISH varied greatly among different mammal species while the organism load was overall low, suggesting a status of colonization or subclinical infection. Severe *Pneumocystis* pneumonia seemed to be very rare. For most of the *Pneumocystis*-positive samples, comparative microscopic examination of H&E- and ISH-stained serial sections revealed an association of the fungus with minor lesions, consistent with an interstitial pneumonia. Colonization or subclinical infection of *Pneumocystis* in the lung might be important in many mammal species because the animals may serve as a reservoir.

## 1. Introduction

*Pneumocystis* spp. are a group of highly diversified opportunistic fungi, which attach to type-1 alveolar epithelial cells [1]. Massive *Pneumocystis* proliferation accompanying severe, often lethal, pneumonia has been described mainly in immunosuppressed humans [2], whereas the infection of immunocompetent hosts is usually asymptomatic and referred to as “colonization” or “subclinical infection” [3]. *Pneumocystis* spp. have been found in the lungs of a wide variety of mammals [1,4]. Members of the mammal orders Artiodactyla [5,6,7,8,9], Carnivora [10,11,12,13], Chiroptera [14], Eulipotyphla [7,15,16], Perissodactyla [17], Primates [18], and the clade Glires consisting of the orders Lagomorpha and Rodentia [19,20] have been investigated during the last decades. Nevertheless, depending on the study focus, different *Pneumocystis* detection methods have been used, making the comparison of data difficult. Histopathological results are especially scarce.

Here we report a large-scale retrospective study aimed to determine *Pneumocystis* prevalence as assessed by in situ hybridization (ISH) in various livestock, pet, laboratory, and wild mammals, and describe characteristic distribution patterns of the fungus in the tissue and the associated histopathology. Additionally, possible factors promoting proliferation of the fungus, such as co-infections and non-infectious concomitant diseases, were analyzed.

## 2. Materials and Methods

We collected formalin-fixed and paraffin-embedded (FFPE) lung tissue samples from 845 mammals involving eight mammal orders (Artiodactyla—even-toed ungulates: *n* = 380, Carnivora—carnivores: *n* = 156, Chiroptera—bats: *n* = 18, Eulipotyphla—insectivores: *n* = 41, Lagomorpha—lagomorphs: *n* = 14, Perissodactyla—odd-toed ungulates: *n* = 36, Primates—primates: *n* = 47, and Rodentia—rodents: *n* = 153). All samples were first screened by ISH for *Pneumocystis* nucleic acid followed by H&E stain for determining histopathological lesions. Details of the host family, species, breed, age, sex, origin, main pathomorphological diagnosis, co-infections, concomitant diseases, and year of sampling are provided in a data repository (https://doi.org/10.34876/13d4-gc84; accessed on 31 January 2023). Except for 82 *Rattus norvegicus*, all animals died or were euthanized and were submitted for routine pathological investigation to the Institute of Pathology or the Research Institute of Wildlife Ecology of the University of Veterinary Medicine Vienna. The owners’ consent for using samples and data were obtained on admission of the cases, and no further ethics permission was required. Eighty-two *Rattus norvegicus* were captured and euthanized as part of the Viennese Rat Project. The capture procedure and method of euthanasia were conducted in accordance with Animal Trial legislation and approved by the institutional ethics and animal welfare committee and the national authority (GZ 68.205/0196-WF/V/3b/2016) on 21 April 2016.

All animals were necropsied according to standard protocols. A set of tissue samples including lungs were fixed in 10% neutral buffered formalin, embedded in paraffin wax (FFPE), sectioned at 2–3 μm, and stained with H&E for assessment of the histological lesions (with details in the data repository https://doi.org/10.34876/13d4-gc84; accessed on 31 January 2023). The lesion scores include score 1 for mild lesions, score 2 for moderate lesions, and score 3 for severe lesions. Grocott’s methenamine-silver nitrate (GMS) staining was used to detect *Pneumocystis* asci.

Chromogenic ISH was performed according to a previously published protocol [21] using an oligonucleotide probe for the 18S rRNA gene conserved in all known *Pneumocystis* species (5′-GGAACCCGAAGACTTTGATTTCTCATAAGATGCCGAGCG-3′) at a final concentration of 20 ng/mL. Hybridization signals were semiquantitated by light microscopy. *Pneumocystis* organism loads were scored as +++ for multiple signals (characterized by almost continuous lining of alveolar spaces over larger areas and frequently completely filling the alveoli), ++ for moderate signals (characterized by either several larger clusters of organisms that fill single alveoli or more diffuse distribution patterns with clusters of organisms predominantly lining the alveolar surface), + for few signals (characterized by the presence of organisms in a few foci, on the surface of alveoli, singly or in small clusters), (+) for minimal signals (only up to five scattered individual organisms), and negative (no organisms in the entire section). The detailed results are provided in the data repository (https://doi.org/10.34876/13d4-gc84; accessed on 31 January 2023).

To confirm the cross-species binding ability of the ISH probe, selected samples from 13 animal species (cattle, sheep, goat, chamois, pig, wild boar, dog, particolored bat, European shrew, European brown hare, rabbit, horse, and rat) were analyzed by PCR and subsequent Sanger sequencing of the *Pneumocystis* mitochondrial small subunit (mtSSU) rRNA gene. Additional attempts to amplify and sequence the *Pneumocystis* mitochondrial large subunit (mtLSU) rRNA gene were not successful for most of these samples (data not shown). Ten FFPE lung tissue sections of 10 μm slice thickness were placed in a 1.5 mL tube, overlaid with 1 mL xylene (Merck, Vienna, Austria), and vortexed. The mixture was incubated for 5 min at room temperature and centrifuged for 5 min at 15,000× *g*. The supernatant was discarded, and the procedure was repeated. After deparaffinization, the pellet was vortexed with 1 mL 100% ethanol (Sigma-Aldrich Handels Gmbh, Vienna, Austria) and centrifuged for 5 min at 15,000× *g*. The supernatant was discarded and the washing step with ethanol was repeated. After centrifugation, the ethanol was removed thoroughly with a pipette. The pellet was dried under vacuum in a desiccator (at least 30 min) and stored at 6 °C for a maximum of 1 day until DNA extraction. DNA was extracted using the QIAmp DNA Micro kit (Qiagen, Vienna, Austria) according to the manufacturer’s instructions and eluted in a 60 μL volume of elution buffer. The PCR mixture consisted of 12.5 μL Kapa 2G Fast HotStart ReadyMix (Sigma-Aldrich Handels Gmbh, Vienna, Austria), 0.4 μM of each primer (SSUF1: 5′-CCT ACA GCT ACC TTA TTT CGA-3′, SSUR1: 5′-ATG RAG TGG GCT ACA GAC GTG A-3′), 1 μL MgCl_2_ (Peqlab, VWR, Vienna, Austria), 2 μL template DNA, and distilled water to a total volume of 25 μL per reaction. The cycler program started with a heat denaturation step at 95 °C for 3 min, followed by 40 cycles at 95 °C for 15 s, 55 °C for 15 s, and 72 °C for 25 s, and terminated with an extension step at 72 °C for 1 min. An aliquot of 10 μL of each PCR product was analyzed by gel electrophoresis using a 2% Tris acetate-EDTA-agarose gel (Sigma-Aldrich, Vienna, Austria). Subsequently, the agarose gel was stained with ROTI^®^ Gel Stain (Roth, Karlsruhe, Germany), and bands were visualized using the GEL DOC™ XR+ gel documentation system (BioRad, Vienna, Austria). Positive PCR products (234–292 bp) were purified using the MinElute PCR Purification kit (Qiagen, Vienna, Austria) and submitted for Sanger DNA sequencing (Microsynth, Vienna, Austria). Sequences were assembled using the software BioEdit Sequence Alignment Editor 7.1.3.0 [22]. The sequences (NCBI accession numbers OP738797-OP738809) were submitted to the NCBI Basic Local Alignment Search Tool (BLAST) and aligned with published mtSSU rRNA sequences of *P. canis* (MT726216 and MT726217), *P. oryctolagi* (NC_060319), *P. carinii* (JX499145), *P. murina* (JX499144), *P*. sp. *ludovicianus* (MT726212), *P*. sp. *macacae* (MT726214), and *P. jirovecii* (JX855936) for confirmation of the affiliation to the genus *Pneumocystis* (Appendix A).

Statistical analyses were carried out with the software IBM SPSS Statistics version 27 (IBM Corporation, Armonk, NY, USA). The association of *Pneumocystis*-positive samples with the age categories “juvenile” and “adult” of the animals was analyzed using the χ^2^ test. Animals were assigned to two categories based on the information at the time of submission. The threshold used was the age of sexual maturity.

## 3. Results

### 3.1. Many Mammal Species Are Colonized with Pneumocystis spp. though Severe Infections Are Rare

A total of 216 out of 845 (26%) investigated samples were *Pneumocystis* positive by ISH. The ability of our ISH probe to cross-detect *Pneumocystis* among the mammal orders Artiodactyla, Carnivora, Chiroptera, Eulipotyphla, Lagomorpha, Perissodactyla, and Rodentia was confirmed by aligning the mtSSU rRNA sequences from 13 selected mammal species (OP738797-OP738809) with published sequences of *P. canis*, *P. oryctolagi*, *P. carinii*, *P. murina*, *P*. sp. *ludovicianus*, *P*. sp. *macacae*, and *P. jirovecii* (Appendix A). The *Pneumocystis* sequences of a dog, a rabbit, and a rat showed 100% nucleotide identity with published sequences. All sequences from hosts for which *Pneumocystis* reference sequences were not yet available also showed matching sequence segments and their affiliation to the genus *Pneumocystis*. The nucleotide sequences of a pig and a wild boar were identical. The mtSSU rRNA sequence derived from a European brown hare differed in only 11 nucleotide positions from *P. oryctolagi*.

Among the mammal orders, Artiodactyla showed the highest *Pneumocystis* prevalence of 38%, followed by Carnivora (33%), Lagomorpha (21%), Perissodactyla (14%), Rodentia (7%), Chiroptera (6%), and Eulipotyphla (2%). None of the samples of the order Primates (*n* = 47) was *Pneumocystis* positive.

At the mammal family level, the highest prevalence of *Pneumocystis* was detected in Canidae (61%) followed by Mustelidae (55%), Suidae (48%), Bovidae (26%), Leporidae (21%), Caviidae (17%), Chinchillidae (17%), Cervidae (14%), Equidae (14%), Cricetidae (10%), Camelidae (9%), Vespertilionidae (9%), Felidae (7%), Muridae (7%), and Soricidae (5%). One of the two members of the family Procyonidae was also positive. Of note, the sample size of the investigated families differed substantially (Table 1).

At the mammal species level, *Pneumocystis* was detected in 36 of 98 investigated mammal species (Table 2), including the following 17 species in which *Pneumocystis* was described for the first time in this study: bison (*Bos bonasus*), blackbuck (*Antilope cervicapra*), chamois (*Rupicapra rupicapra*), water buffalo (*Bubalus bubalis*), alpaca (*Vicugna pacos*), Bactrian camel (*Camelus bactrianus*), gray wolf (*Canis lupus*), Eastern Canadian wolf (*Canis lupus lycaon*), Eurasian badger (*Meles meles*), beach marten (*Martes foina*), Northern American river otter (*Lontra canadensis*), Eurasian river otter (*Lutra lutra*), Oriental small-clawed otter (*Aonyx cinereus*), striped skunk (*Mephitis mephitis*), particolored bat (*Vespertilio murinus*), long-tailed chinchilla (*Chinchilla lanigera*), and black-bellied hamster (*Cricetus cricetus*).

In all mammal species, the number of cases with multiple ISH signals was low (Suidae: 12 domestic pigs (*Sus scrofa domesticus*), 1 wild boar (*Sus scrofa*); Bovidae: 1 sheep (*Ovis aries*), 1 goat (*Capra hircus*); Canidae: 1 dog (*Canis lupus familiaris*); Mustelidae: 1 Eurasian badger (*Meles meles*); Leporidae: 1 rabbit (*Oryctolagus cuniculus*); Muridae: 2 rats (*Rattus norvegicus* and *Rattus rattus*). In pigs, cases with ISH score ++ were the most prevalent while in all other families, cases with ISH scores + and (+) dominated (Table 1).

### 3.2. Pneumocystis Colonizes the Lung with Different Distribution Patterns but the Lung Tissue Is Only Marginally Impaired by the Infection

For most of the *Pneumocystis*-positive samples, comparative microscopic examination of H&E- and ISH-stained lung tissues revealed an association of the fungus either with healthy lung tissue (*n* = 17) or with minor lung lesions consistent with an interstitial pneumonia (*n* = 182). In nine samples, the ISH signals were localized in areas with purulent pneumonia while in eight cases, ISH signals were associated with granulomatous pneumonia.

ISH showed that *Pneumocystis* organisms were primarily located in the alveoli. In lungs with low ISH scores ((+) and +), only a few scattered *Pneumocystis* spp. organisms were attached to the alveolar wall, whereas a continuous lining of alveolar spaces by the organisms was visible in moderately to severely infected lungs (++ and +++). In 22 lung tissue samples, the alveolar spaces were filled with the fungus. However, the distribution pattern of *Pneumocystis* spp. in severely infected pig lungs varied. In some cases, focal large clusters of the fungal spheroids were observed. In the immediate vicinity of these ISH-positive areas, *Pneumocystis*-free lung tissue was present (Figure 1). Diffuse distribution patterns also occurred, and the lungs showed alveolar lining over large lung areas (Figure 2). The lung tissue was either condensed (Figure 2c) or the alveoli were well shaped with no noticeable change in surrounding tissues (Figure 2f). GMS stain revealed the presence of cystic forms of *Pneumocystis*. The comparison of ISH and GMS revealed that more organisms were labeled with ISH, suggesting a larger number of trophic forms in all cases (Figure 1b,e).

ISH on lung tissues from other hosts confirmed the typical distribution patterns with alveolar lining at the beginning of proliferation and a subsequent filling of alveoli in more advanced stages of infection (Figure 3, Figure 4, Figure 5, Figure 6, Figure 7 and Figure 8). The lungs of a laboratory rat (Figure 3a–c) showed a higher fungal load than those of pet rats (Figure 3d,e and Figure 4a,b). In these lungs, an extraordinarily high number of asci was stained by GMS (Figure 3b). Except for two rats with minimal ISH signals, all wild rats were negative. In the lungs of several species, e.g., Eurasian badger (Figure 4c,d), rabbit (Figure 4e,f), and wild boar (Figure 5a–c), the fungus was present in abundance, whereas other mammals, e.g., cat (Figure 6a,b), guinea pig (Figure 6c,d), European shrew (Figure 7a,b), and horse (Figure 7c,d), showed mainly very low fungal loads, suggesting a colonization or subclinical infection.

In 15 animals with high fungal loads, *Pneumocystis* organisms were also detected on the ciliated respiratory epithelium of bronchioles and bronchi (Figure 8 and Figure 9). Single bronchioles of two pigs were completely filled with fungal organisms, and GMS stain confirmed the presence of asci (Figure 8a,b). A mild bronchiolitis was confirmed histologically (Figure 8c). The bronchioles of a severely infected laboratory rat were filled with fluid containing single fungal organisms (Figure 8d,e). In all other samples, only a few scattered fungal organisms were attached to the ciliated respiratory epithelium, and asci could be confirmed by GMS staining (Figure 9a,b). In one sample from a sheep, a significantly higher *Pneumocystis* organism load was observed within the larger respiratory airways than within the alveoli (Figure 9d,e). Most of the bronchioles and bronchi containing fungal organisms showed peribronchial cuffing (Figure 8c,f and Figure 9c,f).

In all mammal species examined, the pulmonary architecture was well preserved, regardless of the severity of the *Pneumocystis* infection. An exception was noted in a Whippet dog whose lung architecture was effaced due to severe *Pneumocystis* pneumonia (PCP). Its alveoli were diffusely filled with many small separated eosinophilic, foamy, and spherical structures in addition to foamy macrophages with variable size (up to 15–25 μm in diameter). The interstitium was moderately infiltrated by lymphocytes and histiocytes. Many small congested interstitial blood vessels had prominent perivascular cuffs of lymphocytes and macrophages. There were also moderate, multifocal areas of hemorrhage throughout the lung and moderate pleural fibrin. ISH or GMS stain confirmed that the fungus was phagocytized by cells resembling macrophages (Figure 10) [23].

### 3.3. Co-Infections and Non-Infectious Concomitant Diseases and Factors

Due to the retrospective nature of the study, evaluation of factors that could potentially positively influence *Pneumocystis* proliferation was difficult. For this reason, the statistical evaluation was limited to the affected age classes (Figure 11). Additionally, a summary of co-infections (Figure 12) and non-infectious concomitant diseases is provided. Animals were assigned to the age categories “juvenile” and “adult” based on the information at the time of submission. The used threshold was the age of sexual maturity. Young animals were significantly more likely to be *Pneumocystis* positive than adults (*p* = 0.009). Eighty percent of the samples with an ISH score of +++ originated from juvenile animals. The percentage of juvenile *Pneumocystis*-positive animals was lower in samples with ISH score ++ (60%), + (58%), and (+) (43%).

Co-infections were documented in 162 of 215 *Pneumocystis*-positive animals. Given the focus of our institute on swine veterinary medicine, most information about other co-infections is available for this species. Nevertheless, further analyses were requested for other species at the time of necropsy. Figure 7 shows the numbers of animals positive for exclusively viral, bacterial, parasitic, or miscellaneous pathogens, and animals positive for co-infections of multiple pathogens. In animals exclusively infected with viral, bacterial, or parasitic pathogens, multiple species of the same categories of pathogens could be detected in the same sample, for example, up to three different viral species or six different bacterial or parasitic pathogens. Co-infections of viral, bacterial, parasitic, and miscellaneous pathogens were detected in pigs (*n* = 70), wild boars (*n* = 4), chamois (*n* = 2), cattle (*n* = 2), sheep, goats, blackbucks, alpacas, oriental small-clawed otters, beach martens, cats, and rabbits (*n* = 1 for each of the latter 8 animal species). The highest number of pathogen species involved in co-infection was 10 in a beach marten. Immunosuppressive pathogens, such as porcine reproductive and respiratory syndrome virus (PRRSV), porcine circovirus 2 (PCV2), classical swine fever virus, canine distemper virus, canine parvovirus, feline panleukopenia virus, feline leukemia virus, mycoplasmas, *Demodex canis*, or *Histoplasma* spp. were either known from prior preliminary reports or detected by necropsy in this study.

Non-infectious concomitant diseases were reported in 19 *Pneumocystis*-positive animals, including different forms of neoplasia (Bactrian camel, dog, and ferret) and diabetes mellitus (dog and ferret).

The details of co-infections and concomitant diseases are provided in the data repository (https://doi.org/10.34876/13d4-gc84; accessed on 31 January 2023).

## 4. Discussion

The objectives of this study were to detect and morphologically describe *Pneumocystis* spp. and associated histopathological changes in the lungs of 845 animals encompassing 8 mammal orders, 31 families, and 98 species, and explore possible factors conducive to the proliferation of this fungus in infected animals. Through this investigation, we demonstrated a highly variable prevalence of *Pneumocystis* spp. among different mammal species, with an overall low *Pneumocystis* load and rare occurrence of severe lung damage. To our knowledge, this is the study of *Pneumocystis* prevalence involving the largest number of animal species, including 17 species described for the first time for the presence of *Pneumocystis* spp.

### 4.1. Pneumocystis spp. in the Artiodactyla Order

Prevalence data obtained to date for *Pneumocystis* spp. in the family Bovidae are limited exclusively to cattle and sheep. Prevalence ranged between 1.6 and 16.8% in cattle from the Czech Republic [24,25], Denmark [5], and Japan [26], and 3.6 and 23% in sheep from Denmark [5] and the Czech Republic [25]. While the prevalence in cattle observed in the present study fell within the published range, we demonstrated a higher *Pneumocystis* prevalence of 37% in sheep. In our study, 5 out of 25 goats were *Pneumocystis* positive. For this species, only two case reports have been published. McConnell et al. (1971) described a case of a four-month-old Boer goat in South Africa with severe diffuse interstitial pneumonia where alveolar spaces were filled by a large number of organisms. Necropsy revealed abomasal and intestinal hemorrhage, which was attributed by the authors to an immune deficiency of unknown etiology [27]. The second case was a three-year-old *Pneumocystis*-positive goat with paratuberculosis. The report does not contain a detailed histological description of fungus or lung lesions [28]. Only one study provides information on the age of investigated cattle and sheep. In the calves and lambs examined, only low *Pneumocystis* loads were detected by cytological examination of lung imprints. No information about concomitant factors or underlying immunosuppressive diseases is provided [5].

Histological studies of PCP in the family Bovidae are very scarce. Kucera et al. (1971) reported organisms filling the alveoli in cattle lung sections, a description indicative of high *Pneumocystis* burden. Apart from the lung of a sheep and a goat with multiple ISH signals in this study, all animals of the Bovidae family showed low fungal loads and minor lung lesions, implying a colonization or subclinical infection. The only sheep with a high *Pneumocystis* load was 9.5 months of age and showed severe bronchopneumonia after infection with *Mannheimia haemolytica*. This bacterium can also act as an opportunist like *Pneumocystis*, gaining access to the lungs when host defenses are compromised by stress or infections with other respiratory pathogens such as viruses or mycoplasmas [29]. The only goat with a high *Pneumocystis* load was juvenile and had bronchopneumonia and endoparasitosis revealed at necropsy. Bacteriological investigation of this animal found low amounts of *Escherichia coli* and *Acinetobacter lwoffii*. The cause of immunosuppression could not be determined from these data, but it cannot be excluded that multifactorial co-infection favored *Pneumocystis* proliferation. In another sheep, a significantly higher *Pneumocystis* organism load was observed on the ciliated respiratory epithelium of bronchioles than within the alveoli. This is a unique finding, and a similar result has not yet been described in the literature. GMS stain proved the presence of asci in the bronchioles and bronchi, which are considered to be the infective stage [30]. Because co-infections with *Streptococcus ovis* and *Mannheimia haemolytica* occurred in this sheep, this histological pattern could reflect an incipient *Pneumocystis* infection. The lungs of other Bovidae studied, including blackbucks, bisons, chamois, and water buffalos, for which there is no report of *Pneumocystis*, showed only weak ISH signals, suggesting a mild infection.

The first description of porcine PCP dates back to the year 1958 in the Czech Republic [31], where *Pneumocystis* cysts were observed in the lungs of piglets with exudative pneumonia. Subsequent studies were mainly concerned with the cytological and histological description of the pathogen and changes in the infected lungs [24,25,32,33,34,35], whereas since the 1980s, factors conducive to *Pneumocystis* proliferation have been increasingly described [5,35,36].

The interplay of respiratory pathogens, collectively known as the porcine respiratory disease complex (PRDC), has long been a concern for researchers. The role of *Pneumocystis* spp. in PRDC was first discussed by Sanches et al. (2006), who postulated a possible association of *Pneumocystis* with PCV2, a potent immunomodulatory pathogen [37]. A rare concurrent infection with *Pneumocystis*, PCV2, PRRSV, *Pasteurella multocida*, and *Streptococcus suis* was described by Kim et al. (2011), who hypothesized a possible synergistic effect between these pathogens as well as an enhancement of *Pneumocystis* infection by the virulence of primary respiratory pathogens [38]. This opinion was shared by Zlotowski et al. (2001), who identified co-infections with PCV2, *Pneumocystis*, *Aspergillus fumigatus*, *Aspergillus flavus*, and *Candida albicans* in wild boars [39], and Borba et al. (2011), who described the concurrent infection of wild boars with PCV2 and *Pneumocystis* [9]. The epidemiological aspect of PCP in pigs has not yet been investigated sufficiently. Esgalhado et al. (2012) suggested asymptomatic carriers as a reservoir for susceptible pigs [40].

The results of the present study supplement and confirm previous findings of our research group. *Pneumocystis* appears to be quite abundant in porcine populations with up to 51% of prevalence reported [41,42,43]. In our study, *Pneumocystis* was primarily located in the alveoli and associated mainly with interstitial pneumonia. The proportion of highly infected cases was higher than that in other animal species. The distribution pattern of *Pneumocystis* in severely infected pig lungs varied substantially, including not only focal large clusters of fungal organisms surrounded by *Pneumocystis*-free lung tissues, but also diffuse distribution patterns with intensive alveolar lining over large lung areas. However, we could not determine a specific reason for these different histopathological distribution patterns. GMS stain proved the presence of asci, but the ratio between cystic and trophic forms differed between pig lungs. In samples with severe infection, *Pneumocystis* organisms were also detected on the ciliated respiratory epithelium of bronchioles, with some bronchi and bronchioles filled with this fungus. Also in these cases, the infective cystic stage was confirmed by GMS staining. Moderate to severe infection occurred mainly in young pigs, especially those co-infected with various other respiratory pathogens, including immunosuppressive pathogens such as PRRSV, PCV2, and mycoplasmas.

Compared to domestic pigs, the *Pneumocystis* prevalence in wild boars was lower (18%). This is not consistent with the prevalence of 50% described by Borba et al. (2011) [9]. In our study, wild boars also showed polymicrobial respiratory infection involving immunocompromising pathogens such as classical swine fever virus.

Information on *Pneumocystis* infections in Camelidae [6] and Cervidae [7] is sparse, and even in our study, only a few animals were weakly positive in ISH. An adult Bactrian camel with an advanced pulmonary adenocarcinoma showed a low *Pneumocystis* load in the lungs.

### 4.2. Pneumocystis spp. in the Carnivora Order

In 1955, Sedlmeier and Dahme published the first report on canine PCP [44]. It occurred in a nine-year-old male German shepherd dog with severe dyspnea, which died two days after admission to the clinic. Histological examination revealed a high load of *Pneumocystis* organisms in the lungs, bronchial lymph nodes, and myocardium. Further cases of spontaneous PCP or latent infections in dogs were published in 1960–1972 [45,46,47]. Farrow et al. (1972) were the first to describe PCP in Miniature Dachshunds though the breed predisposition was not well defined [48]. They described not only acute respiratory symptoms, but also general weakening conditions due to underlying diseases including chronic diarrhea. At that time, the relationship between clinical PCP and immunodeficiency states had already been demonstrated in laboratory animals [49]. Given that five out of the six investigated dogs were closely related, the authors suspected that PCP resulted from congenital or acquired immunodeficiency of the host [48]. During the following years, canine PCP was mainly described in Miniature Dachshunds [50,51,52,53,54,55] and Cavalier King Charles spaniels [56,57,58,59,60,61,62,63]. For both breeds, an impairment of immunity was assumed [52,62]. Secondary PCP as a consequence of X-linked CD40 ligand deficiency was described in a Shih Tzu [64]. Nevertheless, dogs from other breeds without known genetic defect in immunity were also affected by PCP [44,65,66,67,68,69,70,71,72]. In those cases, previous illnesses and chronic co-infections such as demodicosis [51,62,68,69] or canine distemper [70] may have caused the underlying immunodeficiency. In breeds with described congenital immunosuppression, the dogs with PCP are generally young [64,65,72], while there are reports that age is not a crucial factor in dogs with immunosuppression caused by other factors [70,73]. Nevertheless, little information is available for many mammalian species regarding congenital immunosuppression. Even though the incidence of clinical PCP is very low [47,67,74,75], most of the described cases were associated with severe clinical symptoms, and the survival rate was generally low [10]. Also in dogs, *Pneumocystis* carriers without clinical symptoms have to be differentiated from dogs with higher fungal loads and severe dyspnea [67].

In our study, we detected a prevalence of 68% in dogs. As in other mammals, cases with low fungal loads predominated. The earlier case of a Whippet mixed-breed dog [23] was an exception. Unlike all other *Pneumocystis*-positive cases, the lung architecture in this case was effaced and the alveoli were diffusely filled with myriads of spherical organisms. This dog had a history of demodicosis and chronic diarrhea, which was probably the cause of fungal proliferation. Nevertheless, there is no specific explanation for the massive destruction of the lung tissue, and to the best of our knowledge, a comparable case has never been published. Examination of samples from other Canidae revealed only low *Pneumocystis* loads in two wolves in our study. In a molecular-biological study, a prevalence of 46.8% was described in red foxes from Germany [11], whereas an older study only reported a prevalence of 16.7% in Danish red foxes after cytological examination of impression smears [7]. One single red fox was included in the present study, and it was negative for the fungus.

Only a small number of publications are available for *Pneumocystis* in cats. However, cats do not seem to be susceptible to clinical PCP [76]. Immunosuppression itself leads only to mild clinical signs and results in the attachment of only sparse organisms to the alveolar walls [76]. In samples from cats infected with feline leukemia virus, the fungus could not be detected [77]. This is consistent with our results. We could observe *Pneumocystis* spp. only in small amounts in 10.9% of cats, although three of the cats had infections with feline panleukopenia virus, feline leukemia virus, or *Histoplasma* spp., the latter flaring up to disseminated infections due to immunosuppression [78]. Individual organisms attached to the alveolar septa were stainable by ISH. In Denmark, only 2% of cats were *Pneumocystis*-positive in cytological preparations [74], whereas Danesi et al. (2019) published a prevalence of 29% using PCR [12].

In the family Mustelidae, we could detect *Pneumocystis* organisms by ISH in the species beach marten, Eurasian badger, ferret, Eurasian otter, Oriental small-clawed otter, striped skunk, and European mink. Interestingly, ISH revealed that higher fungal loads were present with a diffuse distribution pattern in the lungs of two Eurasian badgers, three ferrets, one Eurasian river otter, and three Oriental small-clawed otters. Samples from these animals were also forwarded for further testing at the time of submission to the Institute of Pathology. One of the Eurasian badgers suffered from canine distemper, two of the Oriental small-clawed otters from enteritis caused by coronavirus and various bacteria, including antibiotic-resistant *Escherichia coli*, and in the lymph nodes of a ferret acid-fast rods were confirmed by Ziehl Neelsen stain. Two case reports have been published so far for representatives of the family Mustelidae. In a mink, severe PCP developed as a result of canine distemper [79]. In a 5-month-old pet ferret, the pathological investigation confirmed a tetralogy of Fallot with a diffuse interstitial pneumonia secondary to *Pneumocystis* spp. [13]. The ferret was seropositive to canine distemper although no evidence of the disease was observed at necropsy [13].

The Procyonidae family was represented by two raccoons, including one with a low *Pneumocystis* load confirmed by ISH. Within the Procyonidae family, there is only one published report about *Pneumocystis* in Brazilian wild white-nosed coatis [80].

### 4.3. Pneumocystis spp. in the Chiroptera, Eulipotyphla, Perissodactyla, and Primates Orders

The detection of *Pneumocystis* in various bat species by PCR is already reported [81,82,83,84,85]. Our present study is the first to describe *Pneumocystis* in the particolored bat with a very low organism load.

Investigation of various mammals belonging to the Eulipotyphla order resulted in a weakly *Pneumocystis*-positive sample from only one European shrew. Representatives of the Soricidae family were generally associated with *Pneumocystis* colonization in the literature, as was the common tree shrew, while Erinaceidae were negative as in our present study [7,18,24,86,87,88,89,90]. There is only one documented *Pneumocystis* case in moles [16].

For the Equidae family, no prevalence data but only single case reports have been published so far. It is currently accepted that *Pneumocystis* remains a significant problem in foals with combined immunodeficiency or other immunosuppressive concomitant factors [17,91,92,93,94,95,96,97,98,99]. The main type of pneumonia in foals was interstitial with hyaline membrane formation [17,91,100]. The *Pneumocystis* load varied from few organisms [96] to multiple organisms [17]. In our study, 29% of the horses showed few ISH signals. Only minor lung lesions were observed. All the horses were younger than two months, of which four were warmblood horses and one a Tinker.

In the present study, we investigated lung tissue samples from 15 different species belonging to the Primates order. All of them were *Pneumocystis* negative. Lungs from primate species in zoos have only been investigated sparsely. A study of these animals from Dutch zoos by impression smear staining reported a *Pneumocystis*-positive result in three Senegal-Galagos, and only one of other species including Demidoff’s-Galago, brown howler monkey, woolly monkey, long-haired spider monkey, white-eared marmoset, and chimpanzee [18]. Further, lung specimens from primates housed in zoos have been used to investigate the phylogeny and evolution of *Pneumocystis* in primates [101,102].

### 4.4. Pneumocystis spp. in the Lagomorpha and Rodentia Orders (Glires Clade)

We investigated lung tissue samples from members of 17 different species of the Glires clade that includes the orders of lagomorphs (rabbit and European brown hare) and rodents (rat, guinea pig, black-bellied hamster, and chinchilla). Only 7% of the investigated rodent samples were *Pneumocystis* positive. In contrast, the prevalence in lagomorphs was higher with 21%. The rats came from different sources: one seven-month-old laboratory rat with severe *Pneumocystis* pneumonia, four pet rats up to five months old with mild to severe ISH grades, and two adult wild urban rats with only scattered individual ISH signals in the lung tissue sections. In the lung of the laboratory rat, the alveoli were filled with spheroid organisms while its bronchioles were filled with exudate containing single asci. The number of asci was extraordinarily high and far exceeded the trophozoite population. Despite the high fungal load, the lung architecture was preserved. In contrast, the lungs of the pet and wild rats showed mainly alveolar lining with fungal organisms and minor lung lesions. Previous studies have shown that the prevalence of *Pneumocystis* in laboratory rats is 80% [103], and that PCP can easily be induced through the application of corticosteroids [104], which has served as a way to establish one of the most commonly used PCP animal models. There are no prevalence data from pet rats. For wild Canadian rats, a prevalence of 47% was reported [19], and for wild Asian rats, it ranged between 57.7 and 82.8% [105,106,107]. The highest prevalence of 92.2% was reported for Danish wild rats [108]. The only reference to the *Pneumocystis* abundance or the severity of pneumonia was from the Canadian study though no organisms were identified in the H&E stained slides [19]. Thus, the infections were probably mild, which is consistent with the findings in our wild rats, even though our prevalence was significantly lower. An explanation for the low *Pneumocystis* prevalence could be that our rats were generally in good health as also any other pathogens could hardly be detected. ISH confirmed a higher fungal load in one pet rabbit and one pet guinea pig. The rabbit was juvenile and had endoparasitosis and bacterial infection. The guinea pig was an adult and had no documented concomitant diseases. There are no prevalence data from pet rabbits, and in pet guinea pigs only one study is available where all three investigated animals were negative [26]. For mice purchased from commercial pet shops, a *Pneumocystis* prevalence of 28.5% has been described [20]; there is no report of *Pneumocystis* in other privately housed rodents. In summary, we found that the prevalence of *Pneumocystis* in the clade Glires was lower compared to published data. A PCR analysis could yield higher prevalence data due to the higher sensitivity compared to ISH. However, besides the estimation of the *Pneumocystis* prevalence in various mammal species, the main focus of this study was to characterize the distribution patterns of the fungus in the lung and its association with histopathological changes in the tissues, an aim only achievable by the combined evaluation of H&E stained and ISH slides. 

## 5. Conclusions

*Pneumocystis* prevalence differed within the mammal orders and was often not comparable to published data, which could be due to different study designs and sample sizes. In all orders, the number of animals with high fungal loads in their lungs was low. Cases with lower *Pneumocystis* loads dominated. Due to the retrospective nature of the study, the provided information on concomitant factors were not consistent and the factors driving the development of severe PCP could not be elicited in all cases. The combination of colonization or subclinical infection of the lung with immunosuppressive conditions might have a relevance for the proliferation of *Pneumocystis* in many mammal species. Furthermore, subclinically infected animals may serve as a reservoir.

## Figures and Tables

**Figure 1 jof-09-00220-f001:**
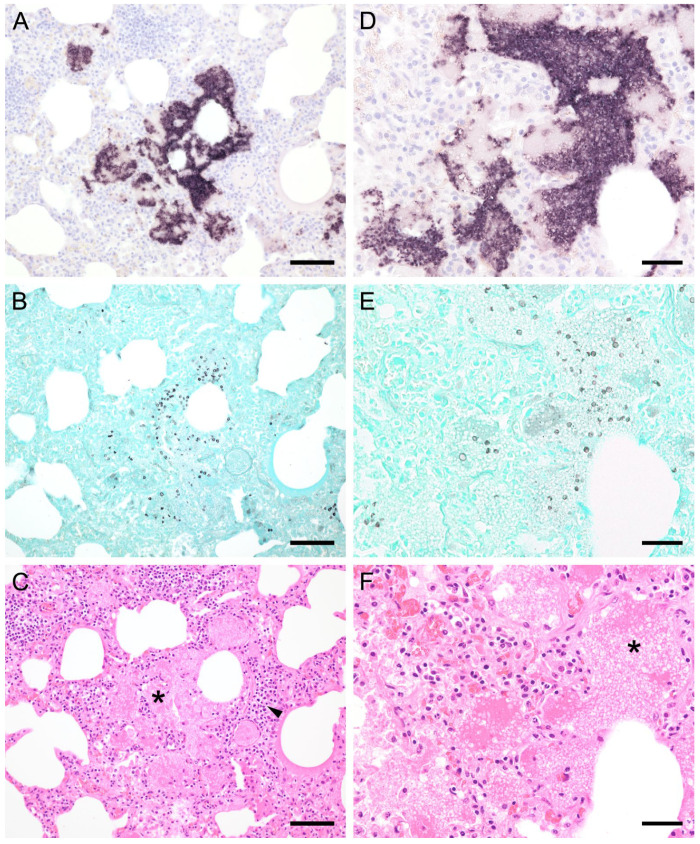
Adjacent sections from the same pig lung area of two different pigs with high loads of *Pneumocystis* spp. organisms are shown in comparison after ISH, GMS, and H&E staining. (**A**) ISH labels large clusters of *Pneumocystis* organisms. The ISH-positive areas are surrounded by *Pneumocystis*-free lung tissue. Bar = 80 μm; (**B**) GMS labels exclusively cystic forms that are less frequent. Bar = 80 μm; (**C**) H&E staining reveals an interstitial pneumonia with moderate lymphocytic infiltration (arrowhead). *Pneumocystis* colonies are recognizable by their ‘honeycomb’ appearance (asterisk). Bar = 80 μm; (**D**) Large parts of the alveolar spaces are filled with ISH-stained organisms due to proliferation of the fungus. Bar = 40 μm; (**E**) The number of GMS-stained asci is small compared to that of unstained trophozoites. Bar = 40 μm; (**F**) Despite the high *Pneumocystis* concentration (asterisk), the pulmonary architecture is well preserved. Bar = 40 μm.

**Figure 2 jof-09-00220-f002:**
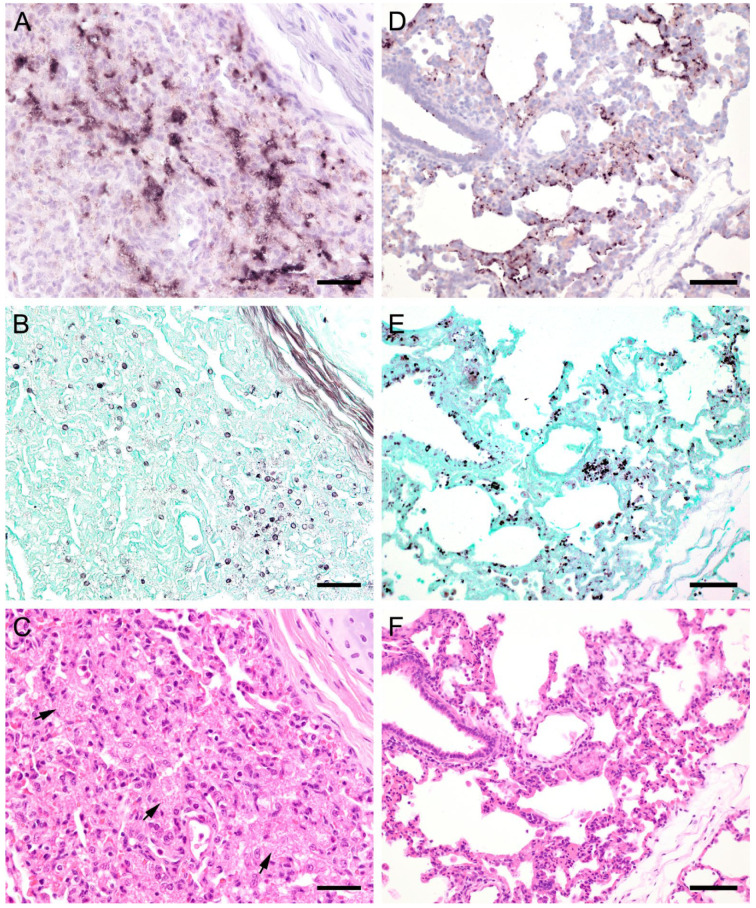
**(see previous page).** Adjacent sections from the same pig lung area of two different pigs with diffuse *Pneumocystis* spp. distribution over large areas are shown in comparison after ISH, GMS, and H&E staining. (**A**) ISH labels *Pneumocystis* organisms attached to the alveolar walls. Bar = 40 μm; (**B**) Compared to the ISH-labeled organisms, the ratio of GMS-labeled cystic forms is higher. Bar = 40 μm; (**C**) H&E staining reveals an interstitial pneumonia, the lung tissue is condensed. *Pneumocystis* organisms are arranged in chains (arrows). Bar = 40 μm; (**D**) ISH reveals alveolar lining with fungal stages over large lung areas. Bar = 80 μm; (**E**) In comparison with ISH, the number of GMS-stained asci is high. Bar = 80 μm; (**F**) The alveoli are well shaped, and the tissue shows only a mild interstitial pneumonia. Bar = 80 μm.

**Figure 3 jof-09-00220-f003:**
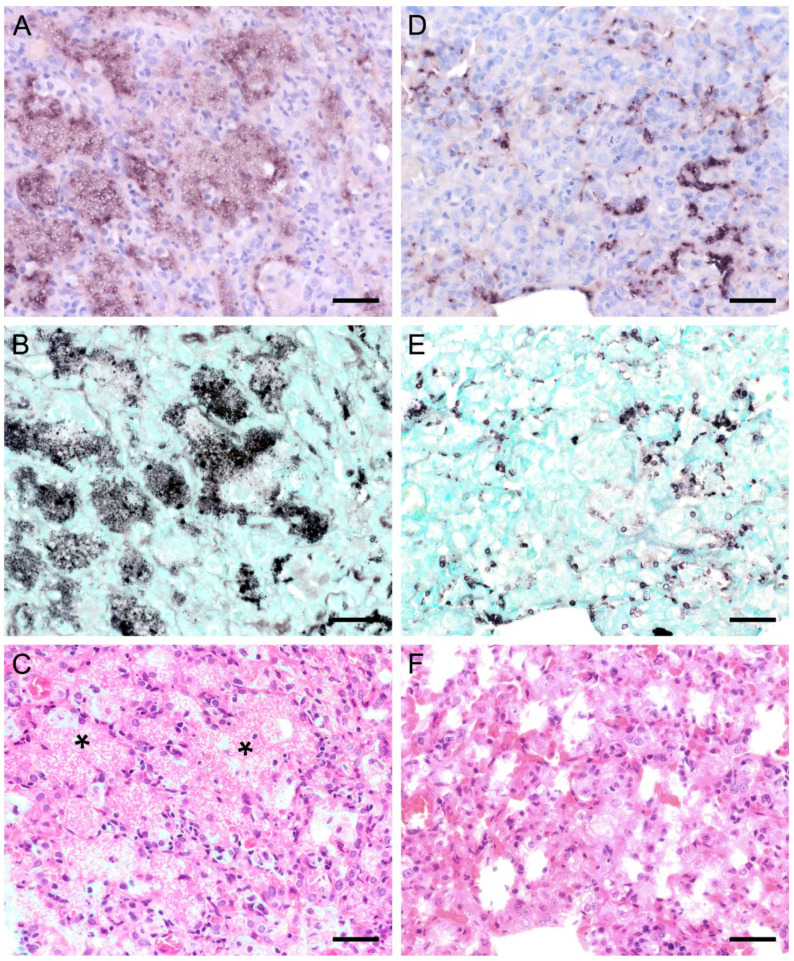
Adjacent sections from the same lung area of a laboratory rat with high and a pet rat with moderate loads of *Pneumocystis* spp. organisms are shown in comparison after ISH, GMS, and H&E staining. (**A**) ISH labels large colonies of numerous *Pneumocystis* developmental stages in the lung of a laboratory rat. The lung alveoli are filled with the fungus. Bar = 40 μm; (**B**) The number of GMS-stained asci is extraordinarily high. Bar = 40 μm; (**C**) H&E staining reveals an interstitial pneumonia with moderate alveolar histiocytosis. *Pneumocystis* colonies are recognizable by their ‘honeycomb’ appearance (asterisks). Bar = 40 μm. (**D**) ISH labels *Pneumocystis* organisms attached to the alveolar walls of the lung of a pet rat. Bar = 40 μm; (**E**) GMS labels exclusively cystic forms Bar = 40 μm. (**F**) H&E staining reveals an interstitial pneumonia Bar = 40 μm.

**Figure 4 jof-09-00220-f004:**
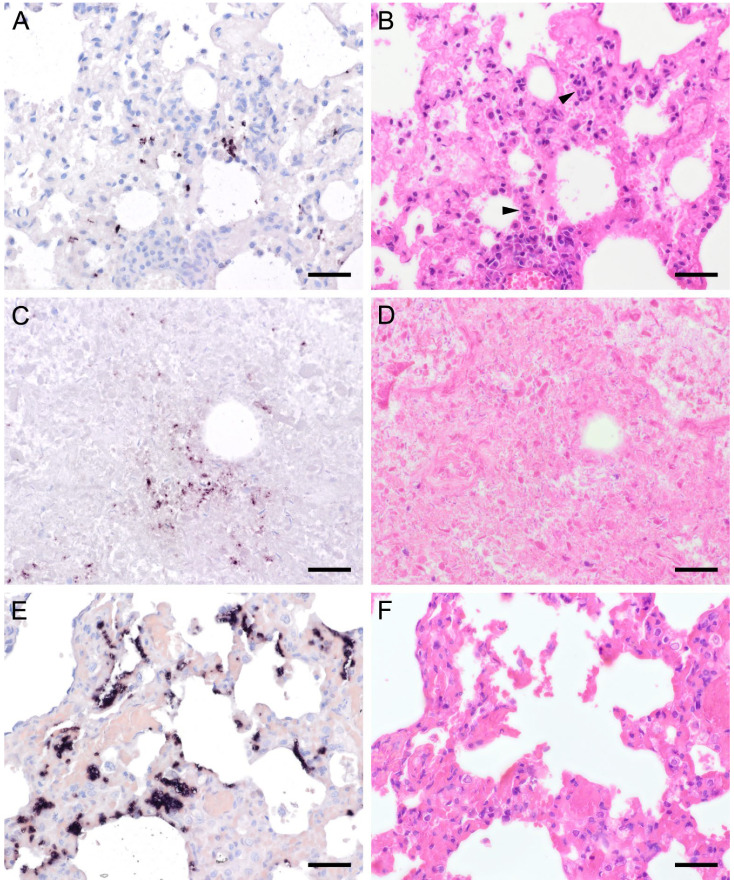
Adjacent sections from the same lung area of a pet rat and a Eurasian badger, each with moderate, and a rabbit with severe colonization with *Pneumocystis* spp. are shown in comparison after ISH and H&E staining. (**A**) ISH signals are diffusely distributed over the entire lung tissue section of a pet rat. Bar = 40 μm. (**B**) H&E staining reveals a granulomatous pneumonia (arrowheads). Bar = 40 μm. (**C**) Diffuse distribution of ISH signals can be seen in several localizations in the lung tissue section of a Eurasian badger. The weak signals are a consequence of autolysis. Bar = 40 μm. (**D**) The state of preservation of the lung is poor due to advanced autolysis. Bar = 40 μm; (**E**) ISH reveals alveolar lining and partly filling of the alveoli of the lungs of a rabbit. Bar = 40 μm; (**F**) H&E staining reveals an interstitial pneumonia. Bar = 40 μm.

**Figure 5 jof-09-00220-f005:**
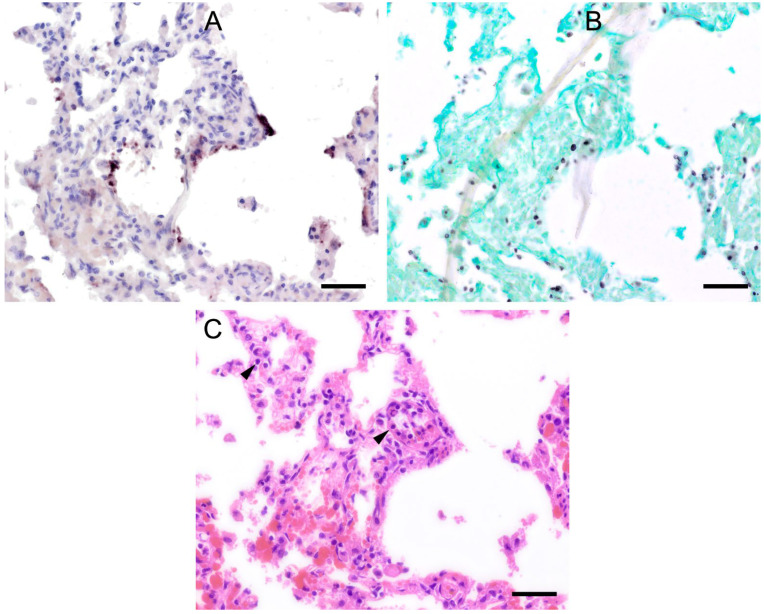
Adjacent sections from the same lung area of a wild boar with moderate colonization with *Pneumocystis* spp. are shown in comparison after ISH, GMS, and H&E staining. (**A**) ISH labels *Pneumocystis* organisms attached to the alveolar walls of the entire lung tissue section. Bar = 40 μm; (**B**) GMS labels exclusively cystic forms. Bar = 40 μm; (**C**) H&E staining reveals an interstitial pneumonia with moderate lymphocytic infiltration (arrowheads). Bar = 40 μm.

**Figure 6 jof-09-00220-f006:**
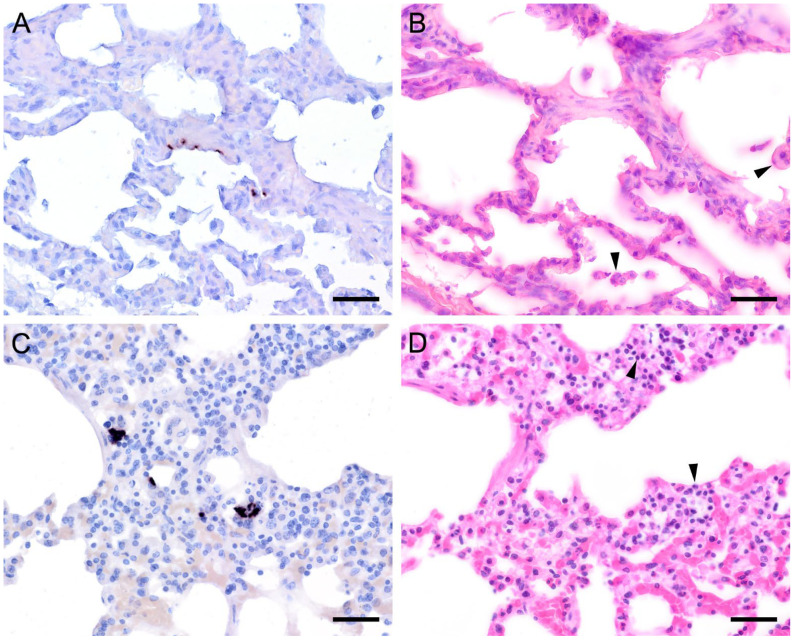
Adjacent sections from the same lung area of a cat and a guinea pig with mild colonization with *Pneumocystis* spp. are shown in comparison after ISH and H&E staining. (**A**) ISH labels single *Pneumocystis* organisms attached to the alveolar walls of the lung of a cat. The rest of the tissue section is *Pneumocystis* negative. Bar = 40 μm; (**B**) H&E staining reveals an interstitial pneumonia with mild alveolar histiocytosis (arrowheads). Bar = 40 μm; (**C**) ISH labels single *Pneumocystis* organisms attached to the alveolar walls of a lung of a guinea pig. Bar = 40 μm; (**D**) H&E staining reveals a granulomatous pneumonia (arrowheads). Bar = 40 μm.

**Figure 7 jof-09-00220-f007:**
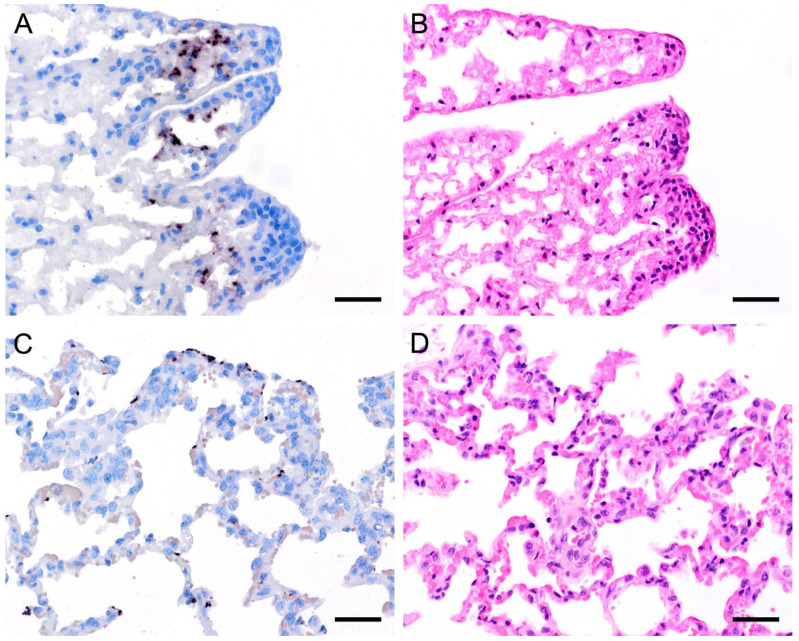
Adjacent sections from the same lung area of a European shrew and horse with mild colonization with *Pneumocystis* spp. are shown in comparison after ISH and H&E staining. (**A**) ISH labels single *Pneumocystis* organisms attached to the alveolar walls of the lungs of a European shrew. The rest of the tissue section is *Pneumocystis* negative. Bar = 40 μm; (**B**) H&E staining reveals an interstitial pneumonia. Bar = 40 μm; (**C**) ISH signals are diffusely distributed over the entire lung tissue section of a horse. Bar = 40 μm; (**D**) H&E staining reveals an interstitial pneumonia. Bar = 40 μm.

**Figure 8 jof-09-00220-f008:**
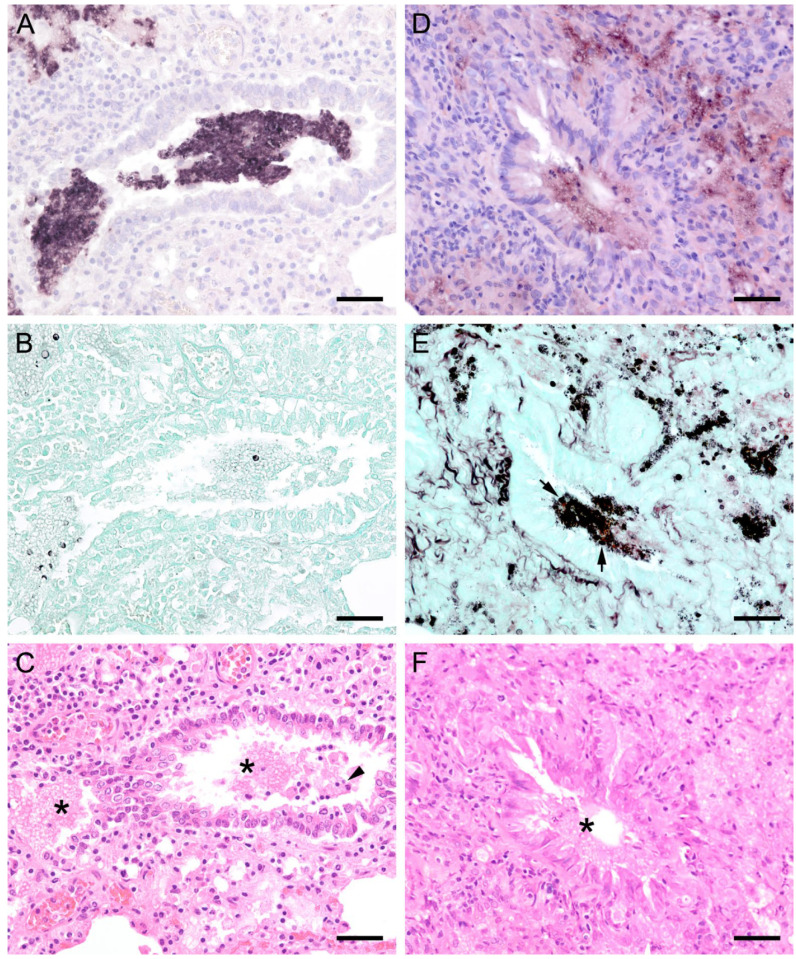
Adjacent sections from the same lung area of a pig and a laboratory rat with *Pneumocystis* organisms in bronchioles are shown in comparison after ISH, GMS, and H&E staining. (**A**) ISH labels a high number of organisms in a bronchiole of a pig. Bar = 40 μm; (**B**) The presence of asci is confirmed by GMS. Bar = 40 μm; (**C**) H&E staining reveals a mild bronchiolitis (arrowhead). The fungal organisms can also be identified histologically (asterisks). Bar = 40 μm; (**D**) ISH labels single organisms in the bronchiole of a laboratory rat. Bar = 40 μm; (**E**) GMS labels single asci (arrows). Bar = 40 μm; (**F**) The bronchiole is filled with fluid and *Pneumocystis* stages (asterisk). Bar = 40 μm.

**Figure 9 jof-09-00220-f009:**
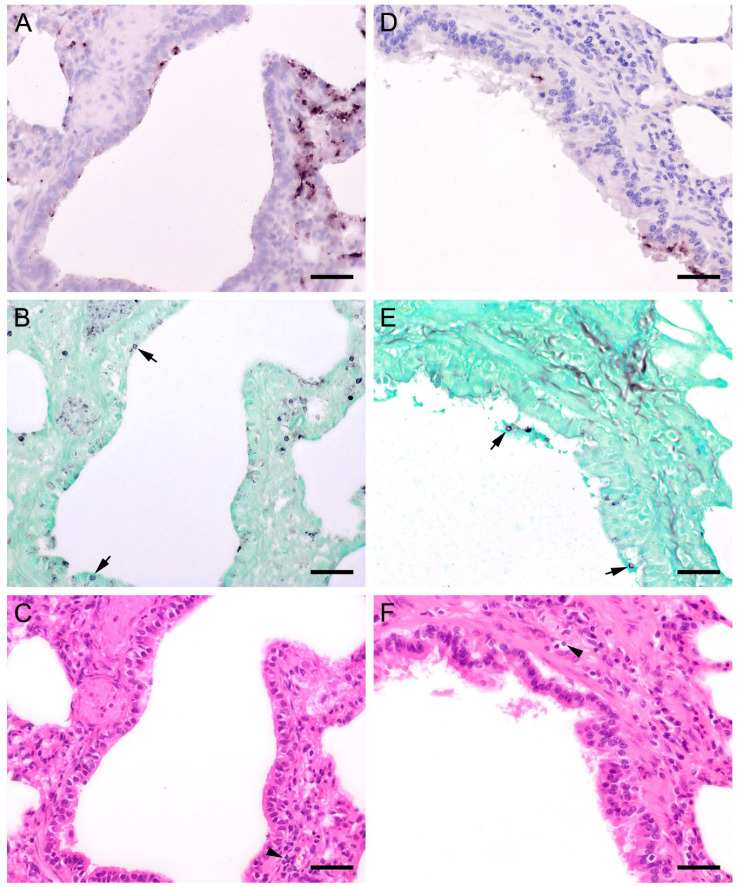
Adjacent sections from the same lung area of a pig and a sheep with *Pneumocystis* organisms in bronchioles are shown in comparison after ISH, GMS, and H&E staining. (**A**) ISH labels single fungal organisms attached to the ciliated respiratory epithelium of a pig. Bar = 40 μm; (**B**) The presence of asci is confirmed by GMS (arrows). Bar = 40 μm; (**C**) H&E staining reveals a peribronchitis (arrowhead). Bar = 40 μm; (**D**) In a lung tissue sample from a sheep, ISH labels *Pneumocystis* organisms within the larger respiratory airways, but not within the alveoli. Bar = 40 μm; (**E**) GMS labels single asci (arrows). Bar = 40 μm; (**F**) H&E staining reveals a peribronchitis (arrowhead). Bar = 40 μm.

**Figure 10 jof-09-00220-f010:**
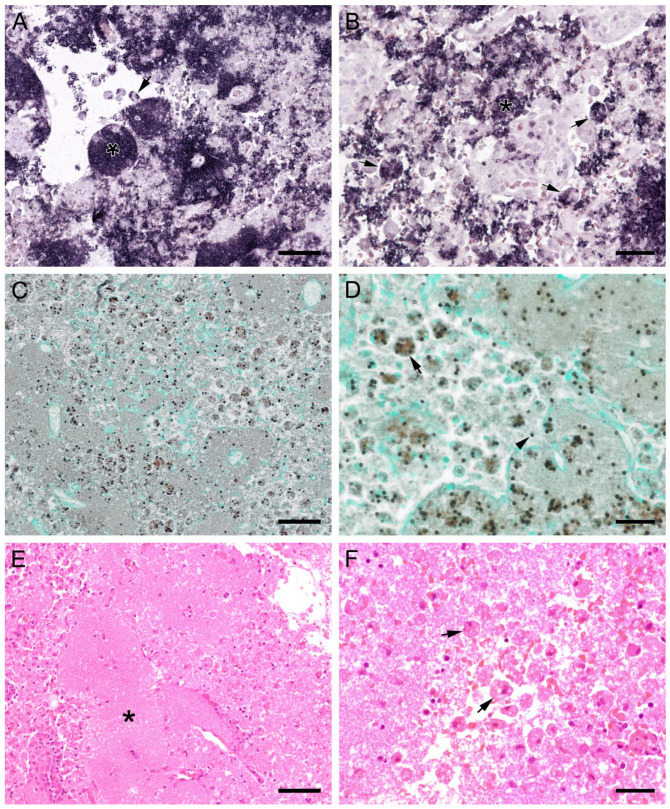
Sections from the lung of a dog with severe PCP. (**A**,**B**) Large areas with numerous developmental stages are labeled by ISH. Signals are present free (asterisks) and within cells with the morphology of alveolar macrophages (arrows). (**A**) Bar = 80 μm; (**B**) Bar = 40 μm; (**C**) GMS labels only cystic stages black. Trophozoites largely remain unstained. Bar = 80 μm; (**D**) Asci are present free (arrowhead) and phagocytized within cells with the morphology of alveolar macrophages (arrow). Bar = 40 μm; (**E**) Severely distended alveolar spaces are diffusely filled with myriads of eosinophilic, foamy, or spherical structures (asterisk). Bar = 80 μm; (**F**) Variably sized cells with the morphology of macrophages containing *Pneumocystis* organisms (arrows). Bar = 40 μm.

**Figure 11 jof-09-00220-f011:**
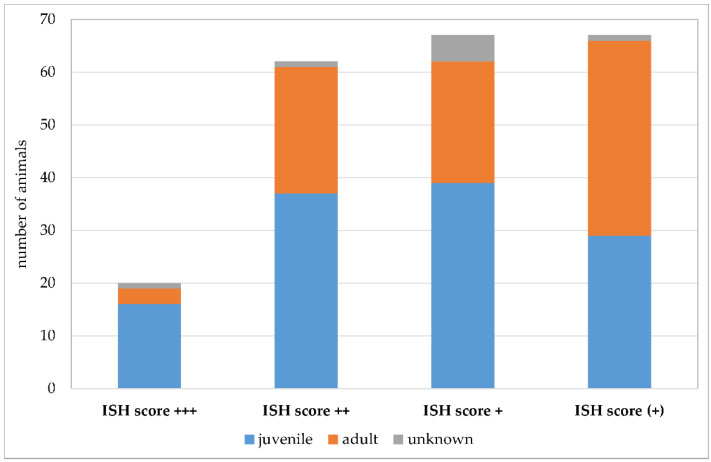
Number of *Pneumocystis*-positive samples according to the ISH scores +++ (multiple signals), ++ (moderate signals), + (few signals), and (+) (minimal signals). Juvenile animals are shown in blue, adult animals in orange, and animals with unknown age in gray.

**Figure 12 jof-09-00220-f012:**
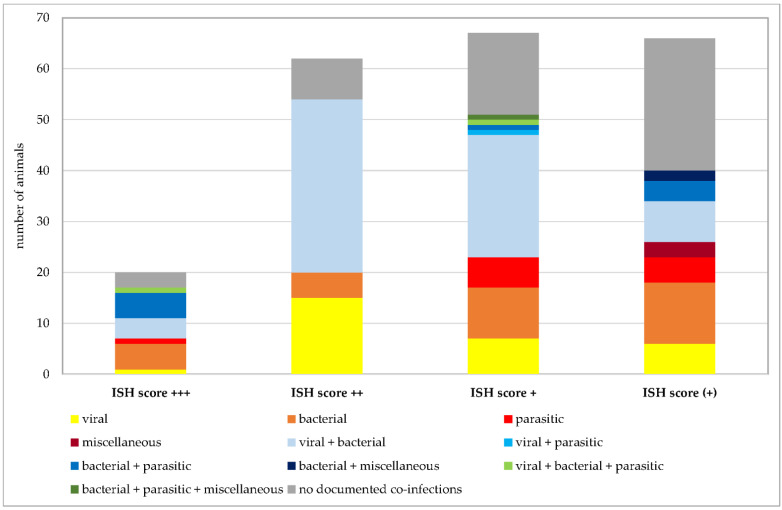
Number of *Pneumocystis*-positive samples broken down to the ISH scores +++ (multiple signals), ++ (moderate signals), + (few signals), and (+) (minimal signals). Animals positive for exclusively viral, bacterial, parasitic, or miscellaneous pathogens are shown in shades of yellow to dark red. Animals positive for combinations of two of these categories are shown in shades of blue and combinations of three of them in shades of green. Animals without documented co-infections are shown in gray.

**Table 1 jof-09-00220-t001:** Number of *Pneumocystis*-positive samples and semiquantitative ISH results for *Pneumocystis* spp. in each mammal family.

Host Order	Host Family	Number of Animals				
		Positive	ISH Score +++	ISH Score ++	ISH Score +	ISH Score (+)
Even-toed ungulates (Artiodactyla) (*n* = 380)	Bovidae (*n* = 96)	25	2	0	10	13
	Camelidae (*n* = 23)	2	0	0	1	1
	Cervidae (*n* = 21)	3	0	0	1	2
	Suidae (*n* = 240)	114	13	51	38	12
Carnivores (Carnivora) (*n* = 156)	Canidae (*n* = 31)	19	1	1	5	12
	Felidae (*n* = 76)	5	0	0	1	4
	Mustelidae (*n* = 47)	26	1	8	5	12
	Procyonidae (*n* = 2)	1	0	0	0	1
Bats (Chiroptera) (*n* = 18)	Pteropodidae (*n* = 6)	0	0	0	0	0
	Rhinolophidae (*n* = 1)	0	0	0	0	0
	Vespertilionidae (*n* = 11)	1	0	0	0	1
Insectivores (Eulipotyphla) (*n* = 41)	Erinaceidae (*n* = 17)	0	0	0	0	0
	Soricidae (*n* = 19)	1	0	0	1	0
	Talpidae (*n* = 3)	0	0	0	0	0
	Tupaiidae (*n* = 2)	0	0	0	0	0
Lagomorphs (Lagomorpha) (*n* = 14)	Leporidae (*n* = 14)	3	1	0	0	2
Odd-toed ungulates (Perissodactyla) (*n* = 36)	Equidae (*n* = 36)	5	0	0	3	2
Primates (Primates) (*n* = 47)	Aotidae (*n* = 1)	0	0	0	0	0
	Callitrichidae (*n* = 19)	0	0	0	0	0
	Cebidae (*n* = 10)	0	0	0	0	0
	Cercopithecidae (*n* = 14)	0	0	0	0	0
	Hominidae (*n* = 2)	0	0	0	0	0
	Pitheciidae (*n* = 1)	0	0	0	0	0
Rodents (Rodentia) (*n* = 153)	Castoridae (*n* = 1)	0	0	0	0	0
	Caviidae (*n* = 12)	2	0	1	0	1
	Chinchillidae (*n* = 6)	1	0	0	0	1
	Cricetidae (*n* = 10)	1	0	0	0	1
	Echimyidae (*n* = 8)	0	0	0	0	0
	Muridae (*n* = 107)	7	2	2	1	2
	Octodontidae (*n* = 4)	0	0	0	0	0
	Sciuridae (*n* = 5)	0	0	0	0	0

ISH score +++: multiple signals, ISH score ++: moderate signals, ISH score +: few signals, ISH score (+): up to five scattered individual signals.

**Table 2 jof-09-00220-t002:** Mammal species found either exclusively positive, negative, or positive/negative for *Pneumocystis* spp. by ISH.

Host Family	Exclusively Positive	Exclusively Negative	Positive or Negative
Bovidae	Chamois (*Rupicapra rupicapra*), water buffalo (*Bubalus bubalis*)		Cattle (*Bos taurus*), sheep (*Ovis aries*), Goat (*Capra hircus*), Blackbuck (*Antilope cervicapra*), Bison (*Bos bonasus*)
Camelidae	Bactrian camel (*Camelus bactrianus*)	Llama (*Lama glama*), Arabian camel (*Camelus dromedarius*)	Alpaca (*Vicugna pacos*)
Cervidae		Red deer (*Cervus elaphus*), Sika deer (*Cervus nippon*), Visayan spotted deer (*Cervus alfredi*), Reindeer (*Rangifer tarandus*), Pere David’s deer (*Elaphurus davidianus*)	Western roe deer (*Capreolus capreolus*), Deer (*Cervus*) ^1^
Suidae			Domestic pig (*Sus scrofa domesticus*), Wild boar (*Sus scrofa*)
Canidae		Red fox (*Vulpes vulpes*)	Dog (*Canis lupus familiaris*), Gray wolf (*Canis lupus*), Eastern Canadian wolf (*Canis lupus lycaon*)
Felidae		Wild cat (*Felis silvestris*), Eurasian lynx (*Lynx lynx*), Lion (*Panthero leo*), Tiger (*Panthera tigris*), Puma (*Puma concolor*), leopard (*Panthera pardus*)	Cat (*Felis catus*)
Mustelidae	Oriental small-clawed otter (*Aonyx cinereus*), Northern American river otter (*Lontra canadensis*), European mink (*Mustela lutreola*)	Marten (*Martes*) ^1^	Beach marten (*Martes foina*), Eurasian badger (*Meles meles*), Ferret (*Mustela putorius furo*), Eurasian river otter (*Lutra lutra*), Striped skunk (*Mephitis mephitis*)
Procyonidae			Racoon (*Procyon lotor*)
Pteropodidae		Indian flying fox (*Pteropus giganteus*), Flying fox (*Pteropus*) ^1^, short-nosed fruit bat (*Cynopterus*) ^1^	
Rhinolophidae		Lesser horseshoe bat (*Rhinolophus hipposidenus*)	
Vespertilionidae		Bat (*Vespertilio*) ^1^, Noctule (*Nyctalus noctula*), Savi’s pipistrelle (*Hypsugo savii*), Common serotine (*Eptesicus serotinus*), Whiskered bat (*Myotis mystacinus*)	Particolored bat (*Vespertilio murinus*)
Erinaceidae		Western European hedgehog (*Erinaceus europaeus*), Middle-African hedgehog (*Atelerix albiventris*), Northern white-breasted hedgehog (*Erinaceus roumanicus*)	
Soricidae	European shrew (*Sorex araneus*)	Bicolored shrew (*Crocidura leucodon*), Lesser white-toothed shrew (*Crocidura suaveolens*), Eurasian pygmy shrew (*Sorex minutus*)	
Talpidae		European mole (*Talpa europaea*)	
Tupaiidae		Common tree shrew (*Tupaia glis*)	
Leporidae			European brown hare (*Lepus europaeus*), Rabbit (*Oryctolagus cuniculus*)
Equidae		Donkey (*Equus asinus*), Pony (*Equus caballus*), Plains zebra (*Equus quagga*)	Horse (*Equus caballus*)
Aotidae		Night monkey (*Aotus trivirgatus*)	
Callitrichidae		Brown-headed tamarin (*Leontocebus fuscicollis*), Cotton-top tamarin (*Saguinus oedipus*), Golden lion tamarin (*Leontopithecus rosalia*), Common marmoset (*Callithrix jacchus*)	
Cebidae		White-faced sapajou (*Cebus capucinus*), Common squirrel monkey (*Saimiri sciureus*)	
Cercopithecidae		King colobus (*Colobus polykomos*), Barbary ape (*Macaca sylvanus*), Gelada (*Theropithecus gelada*), Mantled guereza (*Colobus guereza*), Long-tailed monkey (*Cercopithecus*, *Miopithecus*) ^1^, Rhesus monkey (*Macaca mulatta*)	
Hominidae		Orang-utan (*Pongo*) ^1^	
Pitheciidae		White-faced saki (*Pithecia pithecia*)	
Castoridae		Eurasian beaver (*Castor fiber*)	
Caviidae		Rock cavy (*Kerodon rupestris*)	Guinea pig (*Cavia porcellus*)
Chinchillidae			Long-tailed chinchilla (*Chinchilla lanigera*)
Cricetidae		Dzhungarian hamster (*Phodopus sungorus*), Golden hamster (*Mesocricetus auratus*)	Black-bellied hamster (*Cricetus cricetus*)
Echimyidae		Nutria (*Myocastor coypus*)	
Muridae		Mouse (*Mus musculus*), Spiny mouse (*Mus saxicola*), Mongolian gerbil (*Meriones unguiculatus*)	Rat (*Rattus norvegicus*, *Rattus rattus*)
Octodontidae		Degu (*Octodon degus*)	
Sciuridae		Swinehoe’s striped squirrel (*Tamiops swinhoei*), Eurasian red squirrel (*Sciurus vulgaris*), chipmunk (*Tamias*) ^1^	

^1^ Report did not contain exact species.

## Data Availability

Publicly available datasets were analyzed in this study. This data can be found here: https://doi.org/10.34876/13d4-gc84.

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
