# Peer review of "Detection of Pneumocystis and Morphological Description of Fungal Distribution and Severity of Infection in Thirty-Six Mammal Species"

_jof, 2023, doi:10.3390/jof9020220_

Round 1
Reviewer 1 Report
This article is not complete. I must therefore reject it.
The results are not properly highlighted. It is a descriptive article that does not go into depth in discussing the results.
I would have liked to see the sequences obtained as well as the alignments made and their length. The alignments should have been done with the sequences of the 5 already described Pneumocystis species.
A simple PCR on the samples paraffin-embedded with mtSSU should not have given good results. Furthermore, the marker Mt SSU should have been performed in parallel with MtLSU which is a much more sensitive marker in Pneumocystis animal species.
The literature references are not always adequate and not always up to date, especially when the authors refer to juveniles as a reservoir for pneumocystis. This issue should have been investigated further.
Such low carriage in rodents is very surprising given the bibliography. The same is true for primates where a carriage of 25% has been reported in previous studies. Detection by staining is not good enough to talk about carriage. In primates, for example, only one fingerprint was positive for GMS, whereas there was a carriage of more than 25%.
With regard to bats in the article, it is stated that one bat is positive. However, in Table 1 it appears as negative (Vespertilionidae).
It would be good if the authors could write an article on pigs only, given the number of samples they have, giving more details of their results and giving better arguments.
In conclusion, there are some interesting results but there is far too little discussion of the results compared to what has already been done. Moreover, staining is not very sensitive compared to molecular biology to be able to discuss carriage in the strict sense. The detection of Pneumocystis on paraffin sections is not sufficient, and greatly underestimates the carriage of Penumocystis, this should be discussed further. The article needs to be revised and reoriented.
Author Response
Dear Reviewer 1,
we are thankful for the constructive comments and the accurate review.
Reviewer 1 (R1): This article is not complete. I must therefore reject it.
The results are not properly highlighted. It is a descriptive article that does not go into depth in discussing the results.
Author (A): We appreciate the reviewer’s comments, but we feel that the current paper provides an important addition to the literature by characterizing the morphologic and histopathologic characteristics of Pneumocystis infection in a large number of mammals. The discussion still covers prevalence data compared with other studies, the impact of immunosuppression, the association with interstitial pneumonia, and the significance of subclinical infections. As per the reviewer’s request, we have expanded the discussion to include recent literature and comments on the association between age and congenital immunosuppression in dogs, the impact of other immunosuppressive factors in Canidae, and the difference in sensitivity of PCR and ISH (line 412-416, 504-508, 548-554, 605ff).
R1: I would have liked to see the sequences obtained as well as the alignments made and their length. The alignments should have been done with the sequences of the 5 already described Pneumocystis species.
A: The sequences have been uploaded to NCBI and will be publicly available upon acceptance of the publication. The alignments were actually provided as requested by the reviewer in the initial submission, as supplemental Figure S1, though they were not accessible to the reviewer due to a technical problem. We have appended them to the manuscript to ensure that the reviewer is able to access the figure (line 901ff).
R1: A simple PCR on the samples paraffin-embedded with mtSSU should not have given good results. Furthermore, the marker mtSSU should have been performed in parallel with mtLSU which is a much more sensitive marker in Pneumocystis animal species.
A: In this study PCR was exclusively used for confirmation of the cross-species binding ability of the ISH probe rather than as the primary detection method. The mtSSU primers used in this study were designed from completely conserved regions of the mitochondrial genomes from more than a dozen Pneumocystis species and showed a higher sensitivity than other primers including the published mtLSU primers (Wakefield et al. 1990, 1996) in detecting Pneumocystis in various animal species (Ma et al. unpublished data). In fact, the published mtLSU primers (Wakefield et al. 1990 and 1996), which have been widely used in many studies, contain mismatches to the sequences from multiple Pneumocystis species that potentially affect the efficiency of PCR amplification. We tried to establish also the PCR on the mtLSU gene, which was unfortunately not successful (line 113-114). We sequenced all positive PCR products of mtSSU and verified their sequence identity by comparing the sequences in GenBank and sequences newly obtained from our lab but not yet deposited into public database.
R1: The literature references are not always adequate and not always up to date, especially when the authors refer to juveniles as a reservoir for pneumocystis. This issue should have been investigated further.
A: We have updated the literature references and expanded the discussion. In our manuscript, we do not refer to juveniles as a reservoir for Pneumocystis, but simply indicate that 80% of the samples with an ISH score of +++ originated from juvenile animals. We clarified that in breeds with described congenital immunosuppression, the dogs affected by PCP are generally young. This is not the case in dogs with acquired immunosuppression (line 504-508).
R1: Such low carriage in rodents is very surprising given the bibliography. The same is true for primates where a carriage of 25% has been reported in previous studies. Detection by staining is not good enough to talk about carriage. In primates, for example, only one fingerprint was positive for GMS, whereas there was a carriage of more than 25%.
A: Because a major focus of our study was to provide a morphological description of the fungus and to describe the histopathology associated with infection, rather than to simply describe prevalence based on PCR, we utilized ISH as it allowed us to localize the area of infection and determine if there were associated pathologic changes. Samples that are simply PCR positive would not allow us to evaluate this. We have modified the abstract and introduction to highlight this aim (line 35, line 39, line 61, line 63). Of note, GMS was used only to confirm the presence of asci as an infectious stage, not as a screening tool (line 90-91).
R1: With regard to bats in the article, it is stated that one bat is positive. However, in Table 1 it appears as negative (Vespertilionidae).
A: We appreciate the reviewer identifying this inadvertent error and have corrected the table.
R1: It would be good if the authors could write an article on pigs only, given the number of samples they have, giving more details of their results and giving better arguments.
A: We have included a large number of pigs in the study since a major research focus of our laboratory is the characterization of P. suis infection and it’s clinical significance, and we will be publishing more detailed studies related to this in the future, but we feel it is important to include the current dataset in the manuscript as our goal was to examine the ISH-defined prevalence and associated histopathology in a broad range of mammals.
R1: In conclusion, there are some interesting results but there is far too little discussion of the results compared to what has already been done. Moreover, staining is not very sensitive compared to molecular biology to be able to discuss carriage in the strict sense. The detection of Pneumocystis on paraffin sections is not sufficient, and greatly underestimates the carriage of Pneumocystis, this should be discussed further. The article needs to be revised and reoriented.
A: As noted above, our goal was not primarily to describe the prevalence of Pneumocystis infection, but rather to characterize the morphologic aspects of infection and to describe the histopathology associated with infection, and we have modified the manuscript to reflect this. We recognize that this likely underestimates the carriage rate as defined by PCR, but that was not our primary focus. Nevertheless, we have included this aspect into the discussion (line 613ff).
Reviewer 2 Report
In this study Christiane Weissenbacher-Lang and colleagues described the presence of Pneumocystis in thirty-six mammal species showing fungal distribution and severity of infection by using in situ hybridization (ISH) and hematoxylin and eosin staining for determining histo-pathological lesions.
I congratulate with all authors because the study is excellent, and I really enjoyed reading this manuscript. The study is very well performed. The results are very well described and presented. Conclusions are very interesting.
The only small comment about is not very clear to the reader is the "ISH score (+)" reported in table 1 (column on the right) - that is not very clear to the reader. Could you add some description in the caption if possible please?
Great Job. Kind regards
Author Response
Dear Reviewer 2,
we thank you for the positive review and are happy that our manuscript meets your expectations. Thank you for the constructive comment, which we have of course implemented.
Reviewer 2: The only small comment about is not very clear to the reader is the "ISH score (+)" reported in table 1 (column on the right) - that is not very clear to the reader. Could you add some description in the caption, if possible, please?
Author: As requested, we have added more detail to the criteria of the ISH scores in the Material and Methods section (line 106). We have also expanded the figure legend (line 194-195).
Reviewer 3 Report
I enjoyed this manuscript and found it to be a very important contribution to the Pneumocystis field. Use of the oligonucleotide probe for the 18S probe for the 18S rRNA gene was very specific long with the GMS staining. Confirmation by PCR and Sanger sequencing confirms the data nicely.
Data expands the number of species infected with Pneumocystis, although often times in low numbers.
Author Response
Dear Reviewer 3,
we thank you for the positive review and are happy that our manuscript meets your expectations.